# ARTIFICIAL HARD SUBSTRATE COLONISATION IN THE OFFSHORE HYWIND SCOTLAND PILOT PARK

Rikard Karlsson[1,*], Malin Tivefälth[1,*], Iris Duranović[1,*], Svante Martinsson[1], Ane Kjølhamar [2], Kari Mette Murvoll[2]

[1]Environmental department, MMT Sweden AB, Gothenburg, SE-426 71, Sweden
[2]Equinor ASA, Trondheim, 7053, Norway
*These authors contributed equally to this work

*Correspondence to*: Rikard Karlsson (rikard.karlsson@mmt.se) and Iris Duranović (iris.duranovic@mmt.se)

**Abstract.** Artificial substrates associated with renewable offshore energy infrastructure, such as Floating Offshore Windfarms, enable the establishment of benthic communities with similar taxonomic composition to that of naturally occurring rocky intertidal habitats. The size of the biodiversity impact and the structural changes in benthic habitats will depend on the selected locations. The aim of the study is to assess colonisation, zonation, quantify diversity and abundance, and identify any non-indigenous species present within the wind farm area, as well as to describe changes in the epifouling growth between 2018 and 2020, with regards to coverage and thickness. This article is based on work undertaken within the offshore floating Hywind Scotland Pilot Park, the first floating offshore wind park established in the world, located approximately 25 km east of Peterhead, Scotland. The floating pilot park is situated in water depths of approximately 120 m with a seabed characterised predominantly by sand and gravel substrates with occasional patches of mixed sediments. The study utilised a Work Class Remotely Operated Vehicle with a mounted High Definition video camera, deployed from the survey vessel M/V *Stril Explorer*. A total of 41 structures, as well as their associated subcomponents, including Turbines Substructures, Mooring Lines, Suction Anchors, and Infield Cables, were analysed with regards to diversity, abundance, colonisation, coverage, and zonation. This approach provides comprehensive coverage of whole structures in a safe and time-saving manner. Eleven phyla were observed with a total of 121 different taxa, macrofauna as well as macro- and filamentous algae, identified on the different structures. The submerged turbines measured approximately 80 m in height and exhibited distinct patterns of zonation. Plumose anemone *Metridium senile* and tube building fan worm *Spirobranchus* sp. dominated the bottom and mid-sections (80 m – 20 m) of the turbines while kelp and other Phaeophyceae with blue mussel *Mytilus* spp. dominated top sections of the turbines (20 m – 0 m). A general increase in the coverage of the epifouling growth between 2018 and 2020 was observed, whereas the change in thickness between years was more variable.

## 1 Introduction

The effects on local benthic habitats during installation works and operations of Offshore Wind Farms (OWF) are of complex nature and extend both below and above the surface of the sea. Previous studies have shown that OWFs can impact areas through the introduction and spread of alien species (De Mesel et al., 2015; Wilhelmsson and Malm, 2008), affect organic

matter deposition (De Borger et al., 2021), and carbon assimilation (Mavraki et al., 2020), as well as alter community structures (Coates et al., 2014; Degraer et al., 2020; Hutchison et al., 2020; Wilhelmsson and Malm, 2008) through the loss of soft sediment habitats and the subsequent introduction of artificial hard bottom substrates. The newly created habitat is usually

larger than the lost habitat (Wilson and Elliott, 2009). The recorded impacts also include recovery of the benthic biodiversity as a result of reduced trawling activities (Bergman et al., 2015; Coates et al., 2016) as well as an increase in nurseries for commercially important and/or protected species (Krone et al., 2017). The submerged structures (turbines and subcomponents on the seabed) introduce hard substrates into areas in which there were formerly lacking, thus facilitating colonisation.

Studies conducted at OWFs around the North Sea show that the faunal and floral communities on turbines can further be

categorised into distinct zones from the splash zone to the intertidal and deep subtidal zone (Degraer et al., 2020; De Mesel et al., 2015). These communities tend to develop over time (typically five to six years from the initial settling of organisms to reach the climax stage (Degraer et al., 2020) and evolve in characteristics, progressing from a pioneer stage (years 1 and 2) with sparse colonising taxa to an intermediate stage (years 3 to 5) exhibiting higher diversity followed by the final climax stage (from 6th year and onward) which is dominated by mussels, anemones, and algae. The time taken to reach this final stage is

dependent upon the fundament type (Degraer et al., 2019).

Global primary energy production has seen a 21% increase in consumption between 2009 and 2019, where electricity from renewable sources, as of 2019, comprises 5 % of the total consumed primary energy (BP, 2020). Conventional wind farms are generally confined to shallow coastal waters (<60 m) by technical and engineering constraints. Floating Offshore Wind Farms (FOWF) not being limited by these parameters, open up new possibilities with regards to installation locations.

**1.1 Aim**

Floating Offshore Wind Farms (FOWF), in contrast to most traditional OWFs, are to be located in deeper waters, at greater distances from the coast and other naturally occurring hard bottom habitats not located on the seabed. Therefore, the aim of this study was to 1) Ascertain whether or not similar impacts, with regards to colonisation on turbines and associated structures, to those observed at traditional OWFs were present at the Hywind Scotland Pilot Park, and 2) Assess if any zonation patterns

were present on the Hywind Scotland Pilot Parks structures, similar to those observed at traditional OWFs. 3) To quantify diversity, abundances and 4): identify if any non-indigenous species were present.

**2 Methodology**

**2.1 Study area**

The world's first commercial Floating Offshore Wind Farm (FOWF), The Hywind Scotland Pilot Park, was constructed in

2017 and became operational the same year. The FOWF is located approximately 25 km east of Peterhead on the Scottish east coast and consists of five turbines, located in water depths of 100 m to 130 m. The seabed comprises mainly sand and gravel substrates with mega ripples and occasional boulder fields classified as mixed sediments (Fig. 1).

Unlike conventional, non-floating turbines whose fundaments are secured directly to the seabed, the floating turbines are attached to the seabed using three Suction Anchors attached to the Turbine Substructure by heavy chains. The Turbine

Substructures extend approximately 80 m below the sea surface, acting as a pendulum to keep the structure steady.

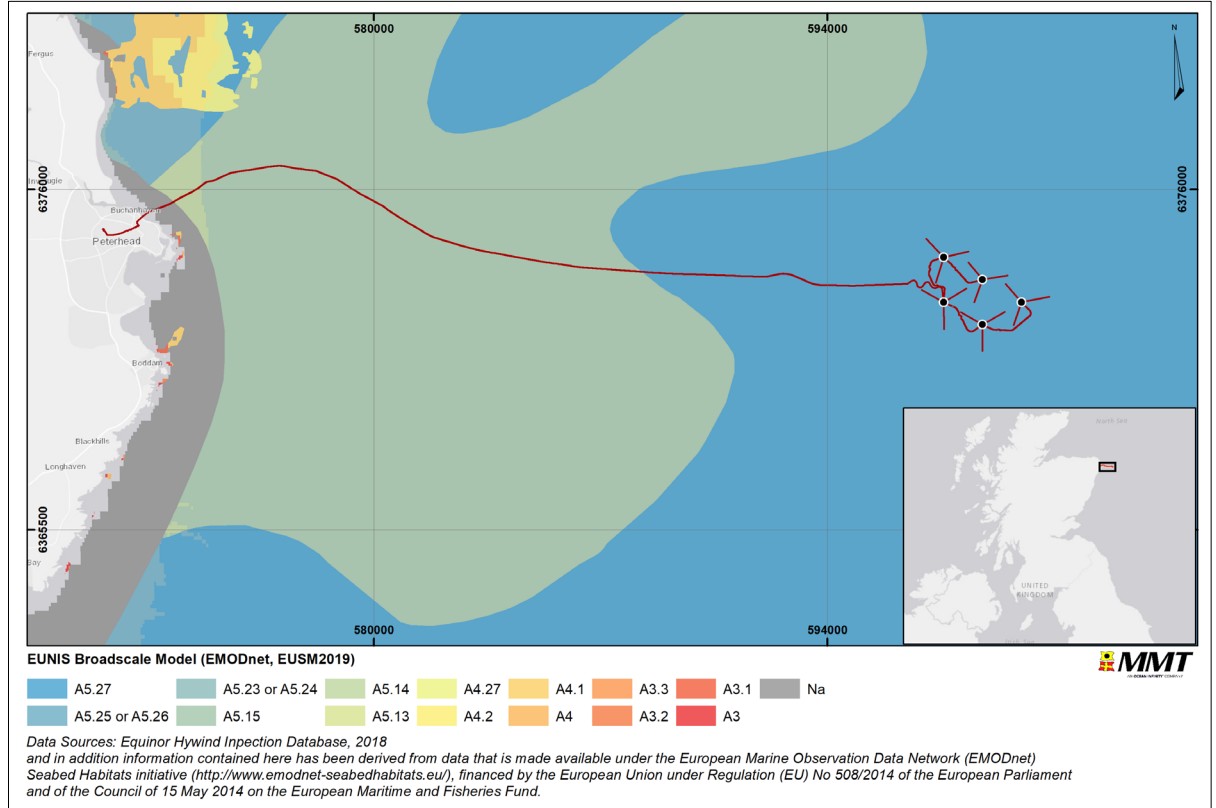

**Figure 1 Overview of the survey area and habitat according to EUNIS classification. The main habitat found in the survey area is A5.27 - Deep circalittoral sand. Other habitats found are: A5.25 – Circalittoral fine sand; A5.26 – Circalittoral muddy sand; A5.23 – Infralittoral fine sand; A5.24 - Infralittoral muddy sand; A5.15 – Deep circalittoral coarse sediment; A5.14 - Circalittoral coarse**
**sediment; A5.13 - Infralittoral coarse sediment; A4.27 – Faunal communities on deep moderate energy circalittoral rock; A4.2 – Atlantic and Mediterranean moderate energy circalittoral rock; A4.1 – Atlantic and Mediterranean high energy circalittoral rock; A4 – Circalittoral rock and other hard substrata; A3.3 - Atlantic and Mediterranean low energy infralittoral rock; A3.2 – Atlantic and Mediterranean moderate energy infralittoral rock; A3.1 - Atlantic and Mediterranean high energy infralittoral rock; A3 - Infralittoral rock and other hard substrata. Basemap sources: © OpenStreetMap contributors 2021. Distributed under the Open**
**Data Commons Open Database License (ODbL) v1.0.**

## 2.2 Data collection

The environmental survey was performed in collaboration with REACH Subsea and occurred simultaneously with a recurring structural inspection of the Hywind Scotland Pilot Park in June 2020. Video footage was obtained using an HD colour camera attached to a Work Class Remotely Operated Vehicle (WROV) supported by LED Flood and Spotlights. Two lasers were

positioned 10 centimetres apart. The WROV maintained a survey speed of 0.3 knots (0.6 km/h). Video footage was recorded during the entire structural inspection of Turbine Substructures, Mooring Lines, Suction Anchors, and Infield Cables (Fig. 2).

Additional video footage, solely for the environmental survey, was collected for Turbine Substructures HS01, HS02, and HS04, Infield Cables HS04 to HS05 (QA01), HS01 to HS04 (QA02), HS02 to HS03 (QA04), and HS03 to HS05 (QA05), as well as the protective Concrete Mattress located on top of the QA01 cable (Fig. 3).

The three priority structures (HS01, HS02, and HS04) were investigated at a reduced speed of 0.2 knots (0.4 km/h), and at three sides (12 o'clock (north), 4 o'clock, and 8 o'clock) of the Turbine Substructures. In contrast, non-priority structures HS03 and HS05 were investigated simultaneously as the structural inspection. The priority structures were investigated from top to bottom at a closer distance compared to the rest of the survey. A distance of approximately 0.5 m was maintained throughout the majority of the environmental survey and areas of interest were investigated at closer distances (<0.3 m). Occasionally,

when sea state or obstructions occurred the distance to the structure was increased up to approximately 1 m. The live feed from the WROV was monitored by one of the marine biologists on shift. This approach allowed for the fauna/areas of interest to be examined in closer detail if required.

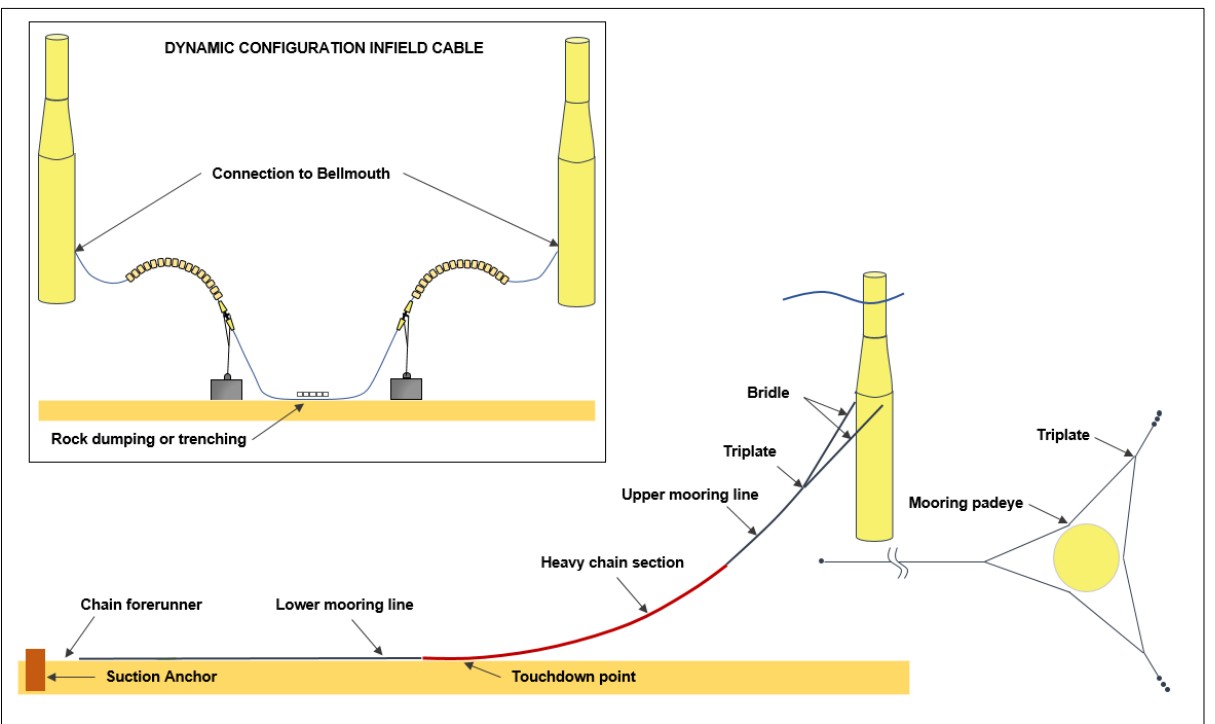

**Figure 2 Layout of Turbine Substructures, Mooring Lines, Suction Anchors and Infield Cables. Figure based on schematic provided by Equinor.**

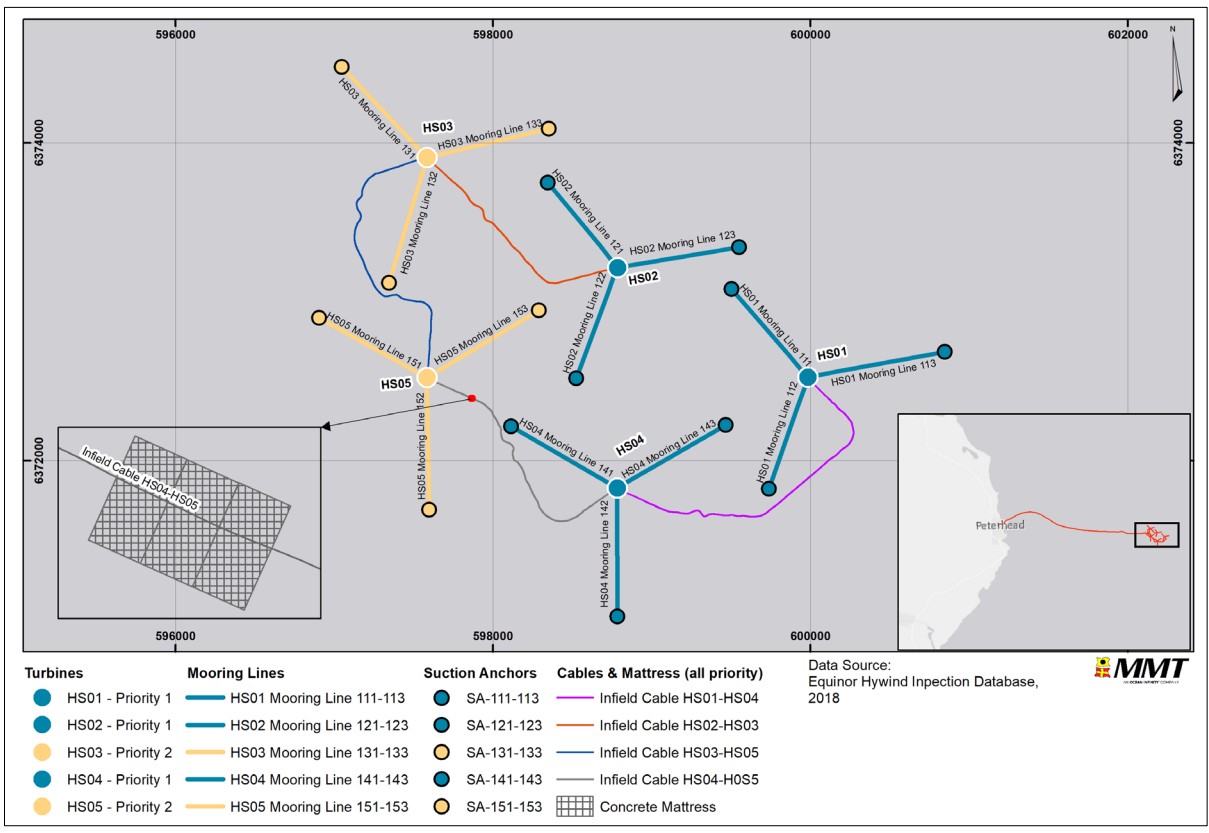

**Figure 3 Overview of survey area and priority and non-priority structures. Basemap sources: © OpenStreetMap contributors 2021. Distributed under the Open Data Commons Open Database License (ODbL) v1.0.**

## 2.3 Analyses methodology

The analyses of the acquired video data were performed in two steps. The first step was analysed in real-time, from the live video feed from the WROV, and included documenting zonation, initial coverage estimates, and common species, which were registered into a field log template in Microsoft Word. During the second step, the video was played back using VLC Media Player and comprised quality control of the field logs as well as enumeration of individuals and assessment of percentage coverage of epifouling species. Lastly, the data was summarised into species lists, with separate lists for each structure and component.

Fauna was identified to the most detailed taxonomic level possible, mainly species, and counted, or noted as present in the case of epifouling faunal (colonial and non-colonial) and floral species. This included the phyla Annelida, Bryozoa, Chlorophyta, Cnidaria, Phaeophyceae, Porifera, and Rhodophyta, as well as for fish, Sessilia, tunicates, and bivalves. When a species could not be identified with certainty, the specimen was grouped into the nearest identifiable taxon of a higher rank, i.e., genus, family, order, etc. Overall coverage of epifouling taxa was quantified, as coverage for individual taxa proved problematic due to different taxa frequently co-habiting on the same spot.

Eggs (from cephalopods, nudibranchs, and gastropods) identified during the survey were excluded from statistical analysis. Asteroidea and sea urchins were occasionally present in such abundance that it was difficult to count each individual, resulting in a likely underestimation of abundance.

### 2.3.1 Additional analyses

Data collected by REACH Subsea during the visual inspections of the structures in October-November 2018 and June 2020 was compiled, and changes in faunal coverage and thickness were compared. The 2018 survey was carried out using similar techniques with the exception of the additional data collected for the environmental survey in 2020, as mentioned in section 2.2. The visual inspection in 2018 was not supported by marine biologists, and species were not recorded but rather growth, shape and, in some cases, phylum/order, whereas the 2020 inspection was aided by marine biologists. To make the two datasets comparable, it was the data collected by the structural inspectors in 2018 and 2020 that were compared.

Known references in the video footage, such as the dimensions of different components, were used to estimate the growth thickness. During the 2020 survey, the addition of parallel lasers spaced 10 cm apart further aided the assessment. Faunal and floral growth was observed for all different components and structures of the wind turbines by REACH Subsea structural inspectors and divided into hard (bivalves, poriferans, barnacles, and tubeworms) and soft growth (bryozoans, hydroids, tunicates, cnidarians, and macroalgae). In this paper, data has been grouped into the three main parts; Turbine Substructures, Mooring Lines, and Suction Anchors, and differences between years were statistically tested using two-tailed paired T-tests in Excel. Structures and subcomponents not reported on during either the 2018 or the 2020 campaign have been excluded in this comparison. In total, 23 turbine sub-components (all included in Turbine Substructures), 125 Mooring Line sections, and 15 Suction Anchors were inspected both years and included in the analyses. Gains and losses of broad groups between the years were noted and used to detect possible succession.

## 3 Results

### 3.1 Identified species

The analyses of data from the Hywind Scotland Pilot Park yielded a total of eleven phyla, with 121 different taxa, 48 taxa were identified to be epifouling fauna and 73 were identified as mobile taxa, in total an estimated number of 15 997 individuals were recorded during the analyses of the survey data (Table 1, Table S1). The most abundant mobile taxon was Asteroidea, likely the common sea star *Asterias rubens*, followed by small sea urchins (*Psammechinus miliaris* and/or *Strongylocentrotus droebachiensis*). Different species of crustaceans were present within the whole survey area and represented the dominating mobile phylum on the seabed. Three possible young colonies of the deep-water coral *Desmophyllum pertusum*, previously *Lophelia pertusa*, were identified along the Infield Cable between Turbines HS01 and HS04. The colony identified at QA02 – HS01 Buoyancy Modules at a depth of 73.5 m (Fig. 4) measured about 20 cm in diameter.

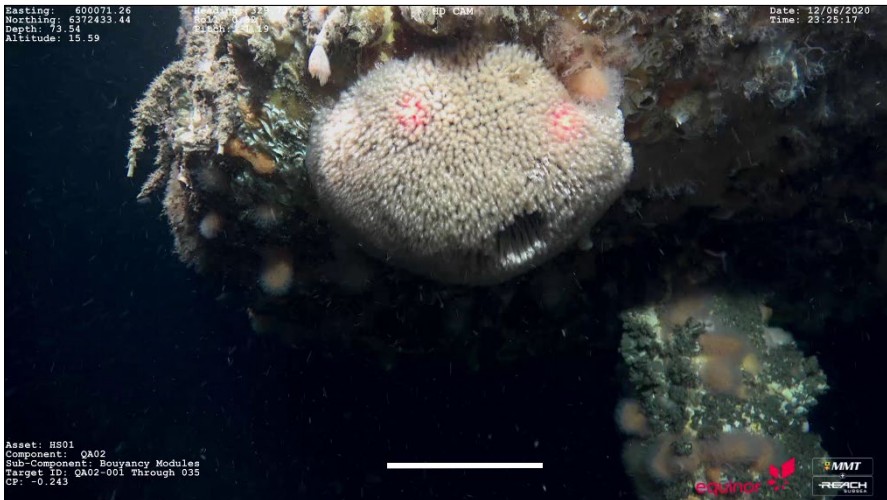

**145** **Figure 4 QA02 – HS01 Buoyancy Modules. Possible young colony of *D. pertusum*. Scale bar = 10 cm.**

No invasive or non-indigenous species were identified during the 2020 survey. However, it should be noted that the use of a WROV without any physical sampling limits the ability to identify smaller species and identify certain filamentous species of red and brown algae.

Species observed on the seabed in close proximity to the structures included different crustaceans (the brown crab *C. pagurus*,
**150** the Norway king crab *L. maja*, different species of squat lobsters, and a few individuals of lobster *Homarus* spp.). Demersal fish, including different species of flatfish Pleuronectiformes, haddock *Melanogrammus aeglefinus,* and ling *M. molva,* were also found in high abundances around the structures. Squids, octopuses, and rays were also observed.

**Table 1 Phyletic composition of fauna and flora identified during visual inspection.**

| Phyla | Number of Epifaunal taxa | Number of Mobile taxa | Number of Individuals of Mobile Fauna |
|---|---|---|---|
| Annelida | 7 | - | - |
| Arthropoda | 1 | 18 | 3 713 |
| Bryozoa | 5 | - | - |
| Chlorophyta | 1 | - | - |
| Chordata | 4 | 28 | - |
| Cnidaria | 21 | - | - |
| Echinodermata | - | 17 | 12 070 (probably underestimated) |
| Mollusca | 1 | 10 | 214 |
| Phaeophyceae | 4 | - | - |
| Porifera | 1 | - | - |
| Rhodophyta | 3 | - | - |
| **Total** | **48** | **73** | **15 997** |

## 3.2 Turbine substructures

The coverage of epifouling taxa was found to be high (~80 % to 100 %), comprising predominantly species *Metridium senile* and *Spirobranchus* sp. across the majority of the turbine surfaces (Fig. 5). The lower intertidal depths were dominated by blue mussels *Mytilus* spp. and brown algae. Mobile taxa present in high abundances included Echinidea, Asteroidea, and Galatheoidea. Squat lobsters were generally noted below 40 m, while grazers such as sea urchins, sea stars, and nudibranchs including *Aeolidia papillosa* were found all over the Turbine Substructures (Fig. 5). Sea urchins and sea stars occurred at all depths but were most abundant between 10 m and 25 m, whereas nudibranchs were more abundant below 40 m.

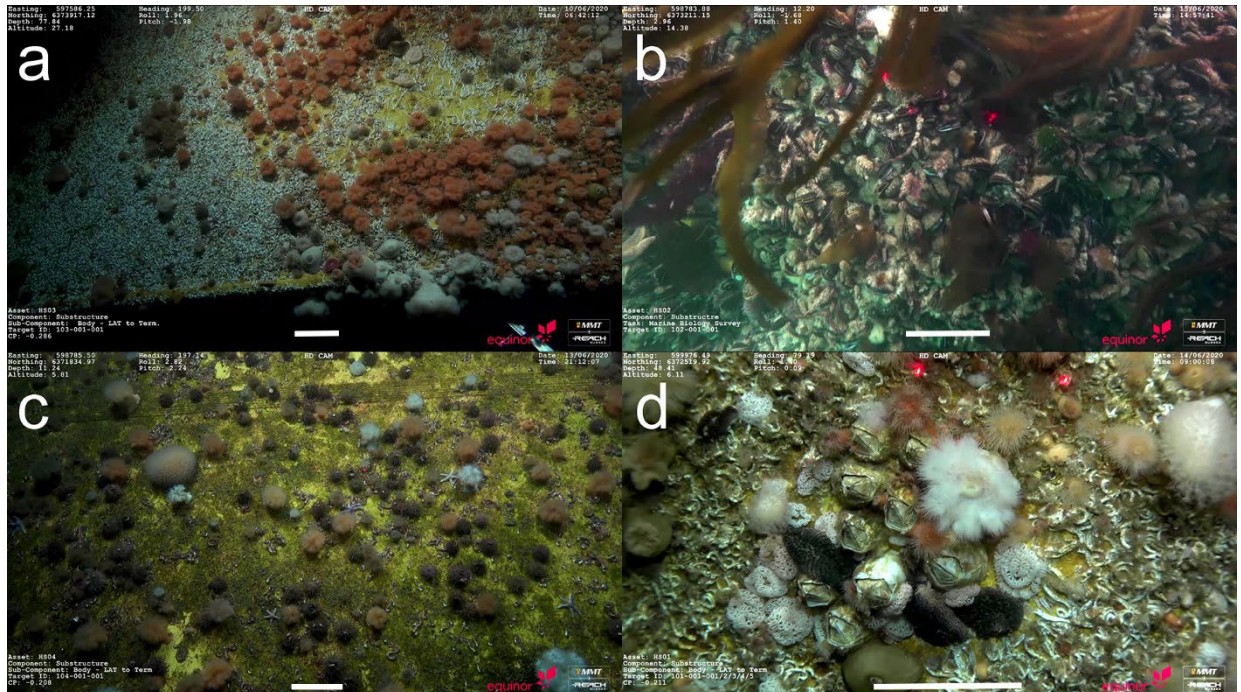

**Figure 5 Example of epifouling colonisation on Turbine Substructures. a.** *Spirobranchus* sp. and *M. senile* at the bottom of HS03 Substructure. **b.** Substructure HS02, with *Mytilus* spp*., Laminaria* sp. and potential amphipod tubes at three m depth. **c.** Substructure HS04, grazing sea urchins and biofilm at 11 m depth. **d.** Substructure HS01, Nudibranch *A. papillosa* and barnacle Balanoidea at 48 m depth. Scale bar = 10 cm.

All Turbine Substructures were further assessed with regards to zonation and faunal composition. The estimated vertical zonation is illustrated in Fig. 6, with the top of the figure representing the sea surface at 0 m extending down to a depth of approximately 77 m representing the bottom of the Turbine Substructure. Four distinct faunal zones were identified at HS01, while HS02 – HS05 comprised five different faunal zones. Turbine Substructure HS01 comprised *M. senile* (50 %) and *Spirobranchus* sp. (50 %) from approximately 30 m to 77 m. At Turbine Substructure HS03, a change in dominating species occurred at approximately 45 m and lower, where *Spirobranchus* sp. was noted to dominate completely. This pattern was also noted for Turbine Substructures HS02, HS04, and HS05 between 60 m to 77 m. Species composition between 4 m and 15 m below the surface differed between the five Turbine Substructures. Turbine Substructure HS01 was colonised by a veneer of biofilm and Phaeophyceae, HS02 by *M. senile* and *Laminaria* sp*.,* HS03 by *Laminaria* sp. and other Phaeophyceae, HS04 by *M. senile*, *Spirobranchus* sp. and biofilm, and HS05 was dominated by *M. senile*, biofilm, and Phaeophyceae. At Turbine Substructure HS01, HS02, and HS03, *Mytilus* spp. and *Laminaria* sp. were the dominating taxa from 0 m to approximately 4 m, and at HS04 and HS05, *Mytilus* spp. and different species of Phaeophyceae were dominant. Potential amphipod tubes could be observed in-between the *Mytilus* spp. located close to the surface.

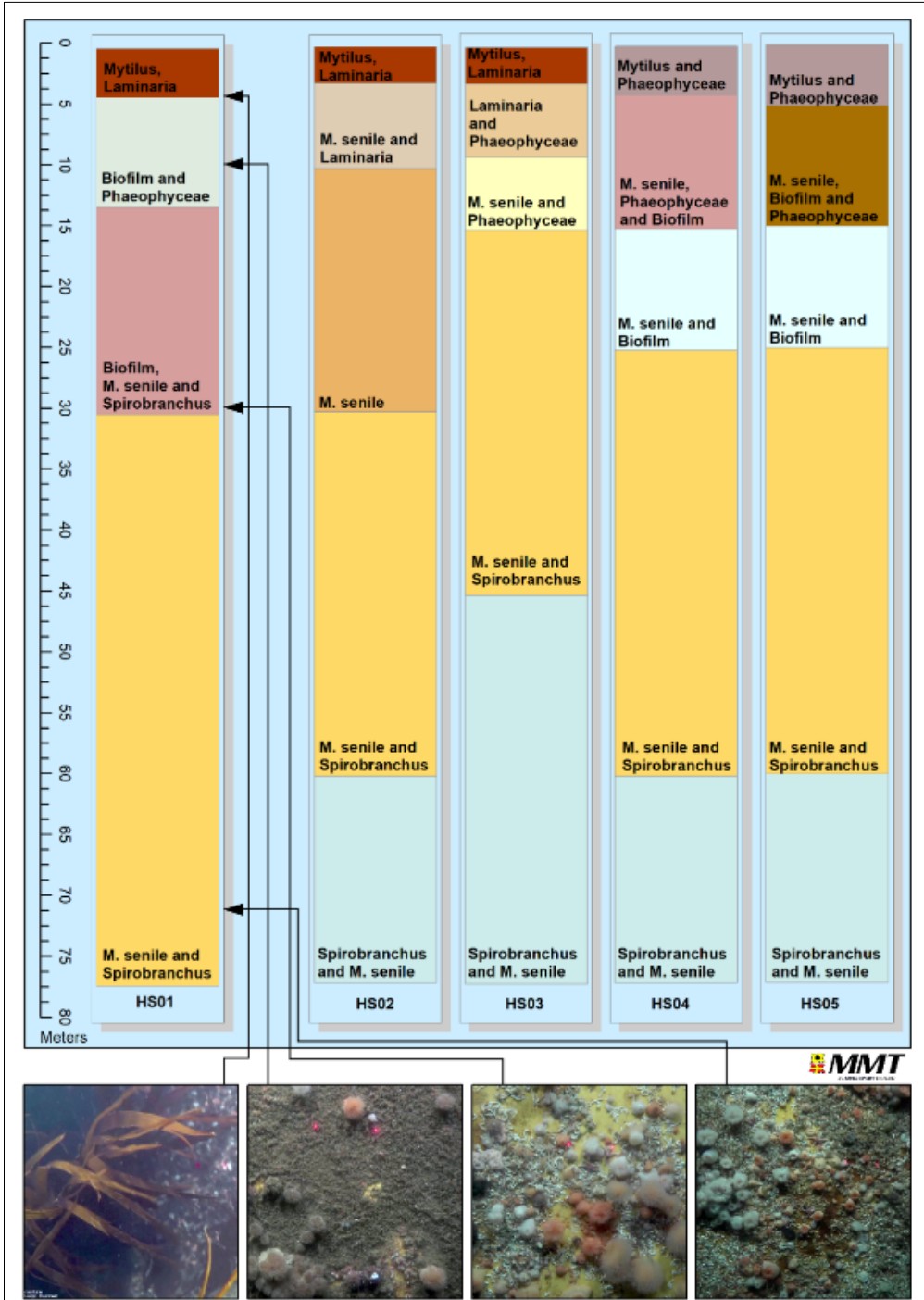

**Figure 6 Illustration of faunal zonation at Turbine Substructure HS01 – HS05. Order of taxa indicates dominance, with dominant taxa listed first.**

### 3.3 Suction anchors

There were no substantial differences between the epifouling communities on Suction Anchors associated with individual Turbine Substructures or between the different turbine groups. Each Suction Anchor was inspected along the top of the structures and separately around the sides. Different hydroids, predominantly *Nemertesia ramosa* and *Ectopleura larynx*, dominated the top of the Suction Anchors with coverage ranging from 20 % to 80 %. *Spirobranchus* sp. and *E. larynx*, with patches of barnacles, dominated the sides of the Suction Anchors with coverage from 60 % to 90 %. Mobile fauna such as

Galatheoidea, *Cancer pagurus*, Palaemonidae, *Lithodes maja,* and nudibranchs were frequently observed.

### 3.4 Mooring lines

No clear differences were noted on the Mooring Lines between the Turbine Substructures, but distinct zonation patterns were observed from top to bottom. The top chain was almost entirely covered by Balanoidea, *M. senile,* and *E. larynx,* with an overall coverage ranging from 60 % to 100 %. The upper-middle chains were similar to the top chains, although the epifouling

decreased as the chains descended towards the seabed with an overall coverage from 40 % to 80 %. The lowest parts of the chains, closest to and on top of the seabed, were dominated by crusts of *Sabellaria spinulosa* and *E. larynx* with coverage ranging from 80 % to 100 %. The Mooring Lines were estimated to have 100 % coverage or close to 100 %, and the composition of the middle chain was similar for all five turbine areas. Mobile fauna found on and adjacent to the Mooring Lines were *A. rubens*, Galathiodea, *C. pagurus*, *L. maja,* and Paguridae. An example of the colonisation along a typical

Mooring Line (Turbine HS01's Mooring Line 111) is presented in Fig. 7, from top to bottom. The top chain was estimated to have an overall coverage between 60 % and 95 %, with an abundance of *M. senile*.

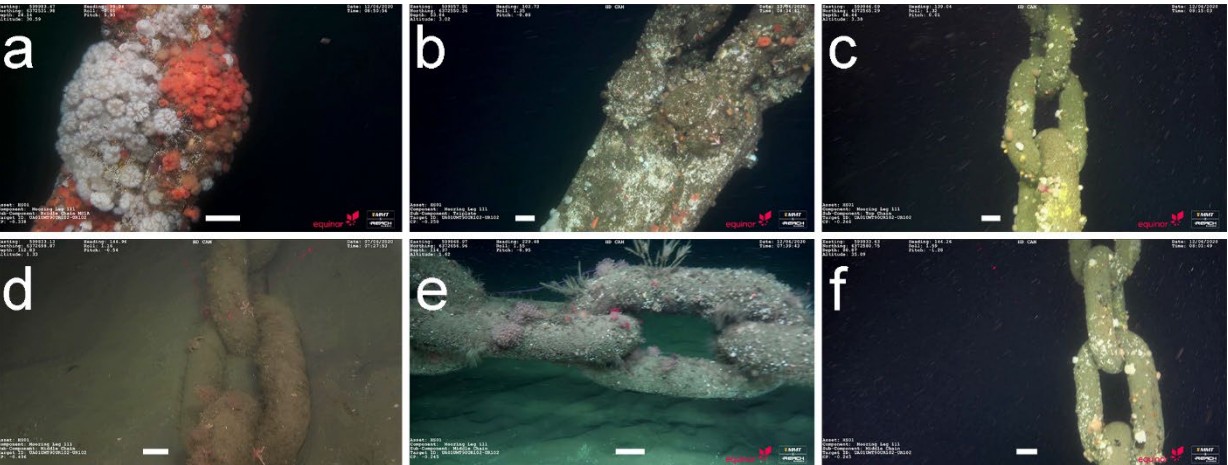

**Figure 7 Example images along a typical Mooring Line (Turbine HS01's Mooring Line 111) top to bottom. a. Top Chain, Bridle Chain. b. Top Chain, Triplate. c. Top Chain. d. Middle Chain, on the seabed. e. Middle Chain, off the seabed. f. Top Chain. Scale**
**bar = 10 cm.**

## 3.5 Infield cables and concrete mattress

From the Bellmouth to Touchdown, the overall dominating species was barnacle Balanoidea, present abundantly along all four Infield Cables. Infield Cables QA01 and QA02 comprised an overall faunal coverage of 100 % from each Bellmouth to Touchdown, whereas QA04 and QA05 comprised areas with lower faunal coverage. The Infield Cables were buried between each touchdown, and no faunal colonisation was therefore present.

The Concrete Mattress, located on top of QA01, was predominantly buried, and the overall faunal coverage was 40 %. The dominating species were *S. spinulosa* and *E. larynx*. Other epifouling fauna present included other hydroids such as *N. ramosa*, *Tubularia indivisa*, and *Urticina* sp. Mobile fauna observed on the structure included Asteroidea, Galatheoidea, Paguridae, *L. maja* and *C. pagurus*. One individual of Pleuronectiformes, *Homarus* sp. and *Molva molva* was present on the Concrete Mattress.

## 3.6 Comparison of faunal growth

Data from the 2018 inspection campaign, provided by REACH Subsea, was compared to the data acquired during the 2020 campaign (Table 2, Fig. 8). The coverage on the Turbine Substructures was not significantly different between the years, neither for the hard (P=0.82) nor for the soft growth (P=0.11). However, there was a significant decrease in the thickness of hard growth (P<0.001), whereas the soft growth increased in thickness (P=0.01). The coverage on the Suction Anchors increased in 2020 compared to 2018, both for the hard growth (P=0.002) and soft growth (<0.001), whereas the thickness of the cover decreased, the change was significant for the hard growth (P<0.001), but not for the soft growth (P=0.10). For the Mooring Lines the coverage increased significantly both for the hard growth (P<0.001) and the soft growth (P<0.001). However, there were no significant changes in the thickness of the growth.

On the Turbine Substructures, the largest shift in composition was a loss of hydroids on 15 of 23 sub-components, and seven sub-components had a gain of macroalgae. On the Mooring Lines, there was a loss of hydroids on 61 of 125 sub-components and a loss of tubeworms on 49 sub-components, and a loss of barnacles on 45 sub-components.

**Table 2 Comparison of mean coverage and thickness of epifouling growth on Turbine substructures, Suction anchors, and Mooring lines between 2018 and 2020.**

| Structure | Growth form | Year | Mean coverage (%) | SD | P | Mean thickness (mm) | SD | P |
|---|---|---|---|---|---|---|---|---|
| Turbine substructure | Hard growth | 2018 | 28.7 | 22.0 | 0.815 | 6.3 | 3.0 | 1.61E-05 |
| | | 2020 | 29.7 | 25.1 | | 2.5 | 0.8 | |
| | Soft growth | 2018 | 60.4 | 27.0 | 0.111 | 35.7 | 33.8 | 0.011 |
| | | 2020 | 69.7 | 22.4 | | 78.3 | 73.2 | |
| | Hard growth | 2018 | 21.0 | 12.4 | 0.002 | 8.7 | 4.6 | 1.04E-04 |

| | | | | | | | | |
|---|---|---|---|---|---|---|---|---|
| Suction anchors | | 2020 | 52.3 | 29.9 | | 23 | 0.8 | |
| | Soft growth | 2018 | 33.0 | 23.4 | 1.06E-05 | 12.3 | 9.6 | 0.10 |
| | | 2020 | 78.0 | 18.2 | | 7.3 | 2.6 | |
| Mooring lines | Hard growth | 2018 | 29.5 | 23.2 | 4.64E-21 | 9.4 | 4.5 | 0.13 |
| | | 2020 | 61.5 | 30.3 | | 12.9 | 25.9 | |
| | Soft growth | 2018 | 55.3 | 24.8 | 2.36E-07 | 22.8 | 12.4 | 0.43 |
| | | 2020 | 71.7 | 24.3 | | 20.6 | 29.3 | |

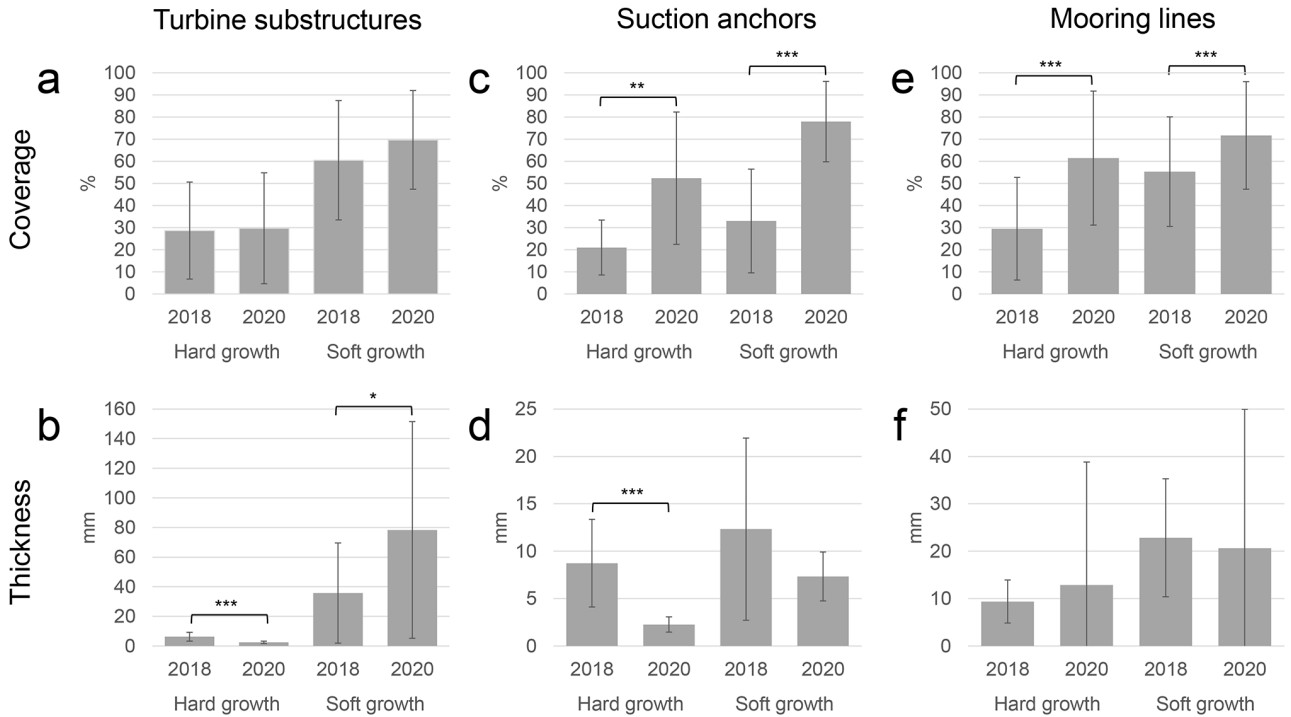

**Figure 8 Coverage and thickness of epifouling growth, shown as mean. Asterisks represent statistically significant differences between years, based on two-tailed paired T-test (\* p < 0.05; \*\* p <0.01; \*\*\* p <0.001). Error bars show ± 1 SD. Note the different scales between b, d, and e.**

## 4 Discussion

### 4.1 Identification of species

The data used in this study was collected from video footage using a WROV. The resolution and quality of the footage limit the detection and identification of smaller organisms, but it is more than sufficient for the detection and identification of larger

organisms. Similar footage has been used successfully in other studies of fauna on offshore structures in the North Sea (e.g., Schutter et al., 2019). However, due to the limit in identifying smaller organisms to a lower level (e.g., species), species diversity and richness will be underestimated (Schutter et al., 2019).

The non-native American lobster, *Homarus americanus*, has been reported from the North Sea and the British islands (Stebbing et al., 2012). Thus, it cannot with certainty be determined whether any of the lobsters observed during the current survey were *H. americanus*. *Homarus gammarus* and *H. americanus* are differentiated morphologically by the absence or presence of spines on the rostrum and are therefore difficult to distinguish without a physical specimen. Hybridisation between these species has also been recorded.

The barnacles observed on the structures were difficult to identify to species level and are grouped in the superfamily Balanoidea. Two possible species have been considered, *Balanus crenatus* and *Chirona hameri*. External experts were consulted and considered *C. hameri* as the probable species, but *B. crenatus* cannot be excluded without a physical sample.

The Mooring Lines and Suction Anchors on the seabed surface have provided additional opportunities for settling and colonisation by *S. spinulosa*, which was identified in the area during previous surveys (MMT, 2013). As the species occurs naturally in the area, the facilitated establishment created by the structures for *S. spinulosa* should not have a negative impact on the habitat. *S. spinulosa* habitats are often associated with high faunal biodiversity (Pearce et al., 2014), which creates feeding grounds for different species of fish.

The shape of the colony tentatively identified as the deep-water coral *D. pertusum* is atypical for the species, however, similar dome-shaped colonies have been recorded on oil platforms in the North Sea (e.g., Gass and Roberts, 2006). Advised experts agree that the colony is likely *D. pertusum,* but due to the small size and uncharacteristic appearance a positive identification would require close up imagery of the calyx using a stills camera. *Desmophylliun pertusum* has not previously been recorded in this area, although colonies have been observed on offshore structures in the North Sea (Roberts, 2002; Bergmark and Jørgensen, 2014). Further, cold-water coral reefs also occur naturally on the continental shelf of western Scotland in water depths of 130 m to 2000 m (Marine Scotland, 2016). Simulations of larval dispersal of *D. pertusum* from offshore structures in the North Sea demonstrate that there is potential for larvae to settle in the survey area (Henry et al., 2018).

## 4.2 Epifouling colonisation and dominant species

The high abundance of *M. senile* is consistent with findings from offshore structures in the North Sea (Whomersley and Picken, 2003; Kerckhof et al., 2012; De Mesel et al., 2015; Kerckhof et al., 2019). Species of the amphipod *Jassa* spp. have previously been identified as one of the dominating species on offshore structures in the North Sea with anemones and hydroids (Lindeboom et al., 2011; Krone et al., 2013) but were not observed during the current survey. The brown matter observed between the blue mussels could be amphipod tubes, such as *Jassa* spp., but a physical sample would be required to confirm this.

The epifouling community differed between the different structures with regard to species diversity. The painted Turbine Substructures harboured fewer taxa compared to the uncoated Mooring Lines. The tube building worm *Spirobranchus* sp. and

the anemone *M. senile* dominated the painted Turbine Substructures while Balanoidea together with hydroids dominated the uncoated structures. Uncoated structures have been noted to comprise more diverse communities than steel monopiles (Kerckhof et al., 2012).

The Concrete Mattress was partially covered by sediment and is likely to be completely buried in the future. The structure provides a hard substrate for epifouling taxa, including Hydroids and *S. spinulosa*. Several mobile taxa were observed, such as lobster, squat lobsters, flatfishes, and ling. Should the structure remain exposed, it could continue to provide a suitable habitat for commercially important species and possibly maintain a *S. spinulosa* reef in the area.

## 4.3 Zonation

A depth zonation similar to, in regard to species composition and distribution, other offshore structures in the North Sea (Whomersley and Picken, 2003; Lengkeek and Bouma, 2009; De Mesel et al., 2015) was noted within the current survey area. Due to safety restrictions concerning close approaches to the Turbine Substructures, estimating the epifouling above the sea surface was not possible. The low intertidal zone was dominated by *Mytilus* spp.*,* which was in line with previous studies conducted in the North Sea (Krone et al., 2013; Bergström et al., 2014). The deep subtidal zone extended from 10 m to 15 metres below the surface and continued down to the bottom. From the low intertidal zone to approximately 25 m depth, there was generally a high presence of biofilm and fewer epifouling species, which could be due to grazing fauna that were occasionally numerous.

Four depth zonations' were observed at Turbine Substructure HS01 and five on Substructures HS02 to HS05. Turbine Substructure HS01 lacked the deepest *Spirobranchus* sp. dominated zonation found at the other four Substructures. The difference is likely due to local variation and faunal spread. The differences were not clear enough to indicate whether or not the currents or the distance to shore would affect the zonation and growth of epifaunal species. The zonation noted along the Mooring Lines comprised a different species community than those identified at the Turbine Substructures. The Mooring Lines were generally dominated by *M. senile* and Balanoidea at the same water depths as where the Turbine Substructures were dominated *by Spirobranchus* sp. and *M. senile*. The top and upper-middle sections of the Mooring Lines were dominated by *M. senile* and Balanoidea. The middle chain comprised, overall, lower faunal colonisation.

## 4.4 Comparison of faunal growth

Coverage of both hard and soft growth has significantly increased from 2018 to 2020 on both Suction Anchors and Mooring Lines, but not on the Turbine Substructures. The change in thickness is more variable compared to coverage, with a significant decrease of hard growth noted on both the Turbine Substructures and Suction Anchors, while an increase of the soft growth thickness was observed on the Turbine Substructures. Large standard deviations were observed for many of the measurements, due to the high variation between the structures. Further, the lack of lasers during the 2018 survey may have contributed to the variation of the measurements between the years.

### 4.5 Succession

The gain and loss of taxa observed indicates a shift in taxonomic composition between 2018 and 2020, with mainly a decrease in hydroids, tubeworms, and barnacles, this was corroborated in discussions with the survey team who performed the initial visual inspection in 2018, and they confirmed that faunal composition had changed between the two years, indicating a succession. The observed changes seem to follow the same trend regarding succession stages that has previously been observed on offshore installations in the North Sea (Rumes et al., 2013; Whomersley and Picken, 2003), tubeworms and hydroids have been reported as the first to colonise the structures, followed by *M. senile* and *Alcyonium digitatum,* who outcompeted the early colonisers by over-growing. This seems to be the case at Hywind FOWF, which would indicate that the park is currently in the species-rich intermediate stage, moving towards a more *M. senile* dominating stage with less biodiversity. The taxonomical resolution in the data collected in 2018 limits the analysis of succession between the years. As in previous studies in the North Sea (De Mesel et al., 2015; Whomersley and Picken, 2003), a zonation was established just a few years after the installation of the structures. Echinoderms were present in high abundance and are considered an important grazer that affects the epifouling community (Witman, 1985) and could keep the epifouling colonisation growth suppressed.

### 5 Conclusion

Species characterisation during visual inspection gave a good overview of the survey area and the higher phyletic community composition. The species detail level was limited when fauna was small and/or the environmental conditions (i.e. strong currents, poor weather, etc.) were poor. To confirm the presence or absence of invasive and non-indigenous species on the structures, physical samples are recommended for future surveys as a complement to the visual inspection. Overall, the approach provides comprehensive coverage of whole structures in a safe and time-saving manner.

The epifouling fauna and flora identified were all species naturally occurring in Scottish waters and around the North Sea. However, the community structure, with its high abundances of *M. senile*, is different when comparing the structures to that which is generally observed on rocky intertidal habitats. *Metridium senile*, *Spirobranchus* sp., *M. edulis,* and barnacles are predominant species typically observed on artificial structures in UK waters and seem to take advantage of newly installed surfaces (Bessel, 2008).

Four mobile taxa featured on the Scottish Biodiversity List and as Priority Marine Features were identified in close proximity of the structures: Atlantic cod *Gadus morhua*, ling *M. molva*, sand eel *Ammodytes* spp., and whiting *Merlangius merlangus*. The overall epifaunal colonisation was assessed to almost 100 % on the different structures, with some minor local variations noted. Epifouling colonisation observed during the survey showed overall similarities with the colonisation of other artificial structures in the North Sea regarding early colonisers and epifouling on structures.

## Data availability

The list of taxa found on the structures is available in supplementary Table S1. The full data set, consisting of video files, is too heavy to upload, but is available upon request.

## Supplement link

Supplementary table S1. List with identified taxa, structure, and quantity.

## Author contribution:

Ane Kjølhamar and Kari Mette Murvoll together funded, conceptualised the survey, and reviewed the manuscript. The survey was carried out by Rikard Karlsson and Malin Tivefälth. Methodology, data analysis and the manuscript draft were equally contributed to by Rikard Karlsson, Malin Tivefälth and Iris Duranovic. Svante Martinsson aided in the revision of the manuscript and performed the statistical analyses.

## Competing interests:

The research was funded by Equinor.

## Acknowledgements

The authors would like to thank Equinor for funding the survey and sharing these findings. The authors would also like to thank the ROV pilots and structure inspectors at REACH Subsea for their assistance as well as the Crew onboard the M/V *Stril Explorer*.

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
