# Peer review of "ARTIFICIAL HARD SUBSTRATE COLONISATION IN THE OFFSHORE HYWIND SCOTLAND PILOT PARK"

_Wind Energy Science, 2021_

## Author Response (AR1)

Dear editor,

We would like to thank the reviewers for their comments and suggestions. We have addressed them accordingly. We revised large sections of the manuscript, including an improved separation of the results and discussion. We have also added statistical testing of the 2018 and 2020 coverage and thickness data and put more emphasis on the observations done within this study. Finally, we have included a species list as a supplementary table. Below are responses to their specific comments, and our changes to the manuscript. As some sections have been revised and split up, we refer to the track changes version of the manuscript for a better overview of the changes. We hope that you will find the manuscript improved and suitable for acceptance for publication in Wind Energy Science.

**Reviewer 1**

1. *I would like to stress the importance of publication of data like the data this paper is based on, please publish the raw data together with the manuscript.*

A species list is now available as a supplementary table.

2. *I would welcome some thoughts in the introduction on why the aims under 1.1 are interesting to study? Do you expect differences between FOWF and standard OWF? Perhaps due to the floating and lack of tidal zonation?*

Additional rationale for the study has been added.

L51-52 "Floating Offshore Wind Farms (FOWF), in contrast to most traditional OWFs, are to be located in deeper waters, at greater distances from the coast and other naturally occurring hard bottom habitats not located on the seabed. Therefore, the" added

3. *I suggest the authors include these newer publications instead of the Vattenfall report. e.g. Bergman, M.J.N., Ubels, S.M., Duineveld, G.C.A., Meesters, E.W.G., 2015. Effects of a 5-year trawling ban on the local benthic community in a wind farm in the Dutch coastal zone. ICES J. Mar. Sci. 72, 962–972. https://doi.org/10.1093/icesjms/fsu193 And some of the papers in the special issue in Oceanography https://tos.org/oceanography/issue/volume-33-issue-4 And the long list of publications from Belgium, such as Degrear et al 2020 instead of the technical report Degraer 2019. Degraer, S., Carey, D., Coolen, J.W.P., Hutchison, Z., Kerckhof, F., Rumes, B., Vanaverbeke, J., 2020. Offshore Wind Farm Artificial Reefs Affect Ecosystem Structure and Functioning: A Synthesis. Oceanography 33, 48–57. https://doi.org/10.5670/oceanog.2020.405*

References have been updated.

L29-49 The introduction now reads: "The effects on local benthic habitats during installation works and operations of Offshore Wind Farms (OWF) are of complex nature and extend both below and above the surface of the sea. Previous studies have shown that OWFs can impact areas through the introduction and spread of alien species (De Mesel et al., 2015; Wilhelmsson and Malm, 2008), affect organic matter deposition (De Borger et al., 2021), and carbon

assimilation (Mavraki et al., 2020), as well as alter community structures (Coates et al., 2014; Degraer et al., 2020; Hutchison et al., 2020; Wilhelmsson and Malm, 2008) through the loss of soft sediment habitats and the subsequent introduction of artificial hard bottom substrates. The newly created habitat is usually larger than the lost habitat (Wilson and Elliott, 2009). The recorded impacts also include recovery of the benthic biodiversity as a result of reduced trawling activities (Bergman et al., 2015; Coates et al., 2016) as well as an increase in nurseries for commercially important and/or protected species (Krone et al., 2017). The submerged structures (turbines and subcomponents on the seabed) introduce hard substrates into areas in which there were formerly lacking, thus facilitating colonisation.

Studies conducted at OWFs around the North Sea show that the faunal and floral communities on turbines can further be categorised into distinct zones from the splash zone to the intertidal and deep subtidal zone (Degraer et al., 2020; De Mesel et al., 2015). These communities tend to develop over time (typically five to six years from the initial settling of organisms to reach the climax stage (Degraer et al., 2020) and evolve in characteristics, progressing from a pioneer stage (years 1 and 2) with sparse colonising taxa to an intermediate stage (years 3 to 5) exhibiting higher diversity followed by the final climax stage (from 6th year and onward) which is dominated by mussels, anemones, and algae. The time taken to reach this final stage is dependent upon the fundament type (Degraer et al., 2019).

Global primary energy production has seen a 21% increase in consumption between 2009 and 2019, where electricity from renewable sources, as of 2019, comprises 5 % of the total consumed primary energy (BP, 2020). Conventional wind farms are generally confined to shallow coastal waters (<60 m) by technical and engineering constraints.  Floating Offshore Wind Farms (FOWF) not being limited by these parameters, open up new possibilities with regards to installation locations."

4. *Please include a definition of all the EUNIS coding used in Fig 1, not only for the blue parts*

The full EUNIS code names are now included in the figure description.

L67-75 the figure legend now reads "**Figure 1 Overview of the survey area and habitat according to EUNIS classification. The main habitat found in the survey area is A5.27 - Deep circalittoral sand. Other habitats found are: A5.25 – Circalittoral fine sand; A5.26 – Circalittoral muddy sand; A5.23 – Infralittoral fine sand; A5.24 - Infralittoral muddy sand; A5.15 – Deep circalittoral coarse sediment; A5.14 - Circalittoral coarse sediment; A5.13 - Infralittoral coarse sediment; A4.27 – Faunal communities on deep moderate energy circalittoral rock; A4.2 – Atlantic and Mediterranean moderate energy circalittoral rock; A4.1 – Atlantic and Mediterranean high energy circalittoral rock; A4 – Circalittoral rock and other hard substrata; A3.3 - Atlantic and Mediterranean low energy infralittoral rock; A3.2 – Atlantic and Mediterranean moderate energy infralittoral rock; A3.1 - Atlantic and Mediterranean high energy infralittoral rock; A3 - Infralittoral rock and other hard substrata. Basemap sources: © OpenStreetMap contributors 2021. Distributed under the Open Data Commons Open Database License (ODbL) v1.0.**"

5. *I am missing when (month + year please) the survey descibed in lines 65 - 75 was carried out and whether this survey was one of the two that were compared in lines 95 - 101.*

Amended.

L78 added "in June 2020"

6. *L58, the anchor chains are connected to the turbine foundation, which is connected to the turbine, the latter I consider to be the large machine on top of the structure, not the whole structure.*

"Turbine" has been revised to state "Turbine Substructure", unless referring to the complete turbine. This change has been implemented throughout the manuscript.

7. *L86 what do you mean by colonisation? The actual colonisation of the foundations by the species was not observed, was it? Perhaps you mean species presence? Of densities?*

Updated to "initial coverage estimate".

L101-103 the sentence reads "The analyses of the acquired video data were performed in two steps. The first step was analysed in real-time, from the live video feed from the WROV, and included documenting zonation, initial coverage estimates, and common species, which were registered into a field log template in Microsoft Word."

8. *L90 I dont understand. In the lines above you state that fauna was counted, but here you state that fauna was scored presence absence. Many Epifouling species can be easily quantified as percentage covered.*

This section has been re-written to be clearer. The coverage of each individual taxa was initially attempted but proved problematic as it was noted that several different taxa (often soft- and hard- growth taxa) occupied the same space, with hard growth taxa such as barnacles and encrusting tube worms occupying spaces in-between anemones. This could generally not be observed but for when, for example, an ROV thruster shifted the anemones' tentacles so that the underlying fauna could be observed.

L107-115 now reads "Fauna was identified to the most detailed taxonomic level possible, mainly species, and counted, or noted as present in the case of epifouling faunal (colonial and non-colonial) and floral species. This included the phyla Annelida, Bryozoa, Chlorophyta, Cnidaria, Phaeophyceae, Porifera, and Rhodophyta, as well as for fish, Sessilia, tunicates, and bivalves. When a species could not be identified with certainty, the specimen was grouped into the nearest identifiable taxon of a higher rank, i.e., genus, family, order, etc. Overall coverage of epifouling taxa was quantified, as coverage for individual taxa proved problematic due to different taxa frequently co-habiting on the same spot.

Eggs (from cephalopods, nudibranchs, and gastropods) identified during the survey were excluded from statistical analysis. Asteroidea and sea urchins were occasionally present in such abundance that it was difficult to count each individual, resulting in a likely underestimation of abundance."

9. *L95 - 101 which month of the year were these surveys carried out? And was it done using similar techniques as the other survey?*

Amended. Yes, it was carried out using similar techniques, and this is now also included in the text.

L118-123 now reads "Data collected by REACH Subsea during the visual inspections of the structures in October-November 2018 and June 2020 was compiled, and changes in faunal coverage and thickness were compared. The 2018 survey was carried out using similar techniques with the exception of the additional data collected for the environmental survey in 2020, as mentioned in section 2.2. The visual inspection in 2018 was not supported by marine biologists, and species were not recorded but rather growth, shape and, in some cases, phylum/order, whereas the 2020 inspection was aided by marine biologists. To make the two datasets comparable, it was the data collected by the structural inspectors in 2018 and 2020 that were compared."

10. *The results only include summary data in Table 1. Please also publish the raw data, e.g. as online supplement, as these type of data are often hard to attain for others and it will increase the citations of your paper*

A species list is now available as a supplementary table.

11. *Fig 5 please explain the difference between 'M senile and spirobranchun' and 'Spirobranchus and M senile'. Is this a dominance difference?*

Yes, the taxa mentioned first was the dominant taxa. This is now stated in the figure legend.

The figure legend now reads "**Figure 6 Illustration of faunal zonation at Turbine Substructure HS01 – HS05. Order of taxa indicates dominance, with dominant taxa listed first.**"

12. *L126 What is biofilm? I know biofilm as microbial layers but I imagine one cannot see this on the footage. I cannot properly see this in Fig 5 as the photo has a very low resolution, but the brown matter between the anemones in the second photo from the left appears as dense aggregations of Jassa spp (aka Jassa silk) or other ampipods. See for example https ://doris.ffessm.fr/var/doris/storage/images/images/jassa_denombrement_my_02/12766088-1-fre-FR/Jassa_denombrement_my_02.jpg*

The text has been revised. The biofilm referred to is the brown veneer often present in the zonation below the Mytilus and Laminaria/Phaeophyceae.

L175-180 section now reads "Turbine Substructure HS01 was colonised by a veneer of biofilm and Phaeophyceae, HS02 by *M. senile* and *Laminaria* sp*.,* HS03 by *Laminaria* sp. and other Phaeophyceae, HS04 by *M. senile*, *Spirobranchus* sp. and biofilm, and HS05 was dominated by *M. senile*, biofilm, and Phaeophyceae. At Turbine Substructure HS01, HS02, and HS03, *Mytilus* spp. and *Laminaria* sp. were the dominating taxa from 0 m to approximately 4 m, and at HS04 and HS05, *Mytilus* spp. and different species of Phaeophyceae were dominant. Potential amphipod tubes could be observed in-between the *Mytilus* spp. located close to the surface."

13. *L143 How did you test for significance of differences between the mooring lines? This is not described in the methods.*

"Significant" has been replaced by "clear".

L192-193 the sentence reads "No clear differences were noted on the Mooring Lines between the Turbine Substructures, but distinct zonation patterns were observed from top to bottom."

14. *Figure 6 please include a size reference or a statement on the size of the chain links*

Scale bars have been added to appropriate figures.

15. *L170 I dont understand, the overall change in faunal thickness has decreased? What does this mean? Did you mean that there was a decrease in thickness between the 2 years? So the change is negative?*

Yes, the thickness of growth has decreased for some of the structures. The section has been updated so to included statistical tests.

L218-228 The section now reads "Data from the 2018 inspection campaign, provided by REACH Subsea, was compared to the data acquired during the 2020 campaign (Table 2, Fig. 8). The coverage on the Turbine Substructures was not significantly different between the years, neither for the hard (P=0.82) nor for the soft growth (P=0.11). However, there was a significant decrease in the thickness of hard growth (P<0.001), whereas the soft growth increased in thickness (P=0.01). The coverage on the Suction Anchors increased in 2020 compared to 2018, both for the hard growth (P=0.002) and soft growth (<0.001), whereas the thickness of the cover decreased, the change was significant for the hard growth (P<0.001), but not for the soft growth (P=0.10). For the Mooring Lines the coverage increased significantly both for the hard growth (P<0.001) and the soft growth (P<0.001). However, there were no significant changes in the thickness of the growth.

On the Turbine Substructures, the largest shift in composition was a loss of hydroids on 15 of 23 sub-components, and seven sub-components had a gain of macroalgae. On the Mooring Lines, there was a loss of hydroids on 61 of 125 sub-components and a loss of tubeworms on 49 sub-components, and a loss of barnacles on 45 sub-components."

16. *The discussion presents several results that are not in the results section. These should be moved to results and only discussed, not presented (again).*

Amended.

17. *L176 The starting statement is a result and should be presented there*

Amended.

18. *L190 need a reference for your statement that S. spinulosa habitats are often associated with high faunal biodiversity*

A reference has been added.

L255-256 the sentence now reads "*S. spinulosa* habitats are often associated with high faunal biodiversity (Pearce et al., 2014), which creates feeding grounds for different species of fish."

19. *L192 need a reference for 'which could further benefit commercial fish species' and please elaborate on the benefit, how does this benefit commercial species?*

The sentence has been removed.

20. *L195 you are introducing new results in the discussion, these should be included in the results section, together with fig 9. At what depth was this recorded and what was the size?*

The section has been split up, and parts have been moved to the Results section. The size of the colony (~20 cm) and the depth (73.5 m) have been added to the text.

L141-143 the section in the results reads as "Three possible young colonies of the deep-water coral *Desmophyllum pertusum*, previously *Lophelia pertusa*, were identified along the Infield Cable between Turbines HS01 and HS04. The colony identified at QA02 – HS01 Buoyancy Modules at a depth of 73.5 m (Fig. 4) measured about 20 cm in diameter."

21. *Fig 9 where in the picture is the D pertusum colony? This round shape in the top middle? Indeed an atypical and very closed growth form. Please include a size reference*

Yes, the shape is atypical, but similarly shaped colonies have been observed on oil platforms. The colony is about 20 cm in diameter, this is now added to the text. There are two lasers, 10 cm apart in the figure, but they are unfortunately hard to see. A scale bar has been added to the figure.

22. *L205 - 207 some of these are not stated in the results, this should go to results section.*

Amended.

23. *L225 I dont understand how a floating structure can have an intertidal zone. Since it is floating the water surface is always at the same position on the structure, is it not?*

The structures are anchored to the seabed using suction anchors and are attached to these by heavy chains. This results in an intertidal zone. However, the intertidal zone is likely reduced as compared to a non-floating structure.

24. *L229 I am confused about teh intertidal zones in this line. What is the difference between low intertidal and deep intertidal?*

The sentence has been amended.

L287-289 the sentence now reads "From the low intertidal zone to approximately 25 m depth, there was generally a high presence of biofilm and fewer epifouling species, which could be due to grazing fauna that were occasionally numerous."

*25. L233 a difference is either significant or not, it cannot be significant enough. Probably you meant large enough?*

Amended.

L292-293 the sentence now reads "The differences were not clear enough to indicate whether or not the currents or the distance to shore would affect the zonation and growth of epifaunal species"

*26. L238 - 245 is a lot of information that should be in the results, new data is not expected in the discussion*

The text has been revised and split up between the Results and Discussion sections.

L226-228 reads: On the Turbine Substructures, the largest shift in composition was a loss of hydroids on 15 of 23 sub-components, and seven sub-components had a gain of macroalgae. On the Mooring Lines, there was a loss of hydroids on 61 of 125 sub-components and a loss of tubeworms on 49 sub-components, and a loss of barnacles on 45 sub-components.

L306-317 reads: The gain and loss of taxa observed indicates a shift in taxonomic composition between 2018 and 2020, with mainly a decrease in hydroids, tubeworms, and barnacles, this was corroborated in discussions with the survey team who performed the initial visual inspection in 2018, and they confirmed that faunal composition had changed between the two years, indicating a succession. The observed changes seem to follow the same trend regarding succession stages that has previously been observed on offshore installations in the North Sea (Rumes et al., 2013; Whomersley and Picken, 2003), tubeworms and hydroids have been reported as the first to colonise the structures, followed by *M. senile* and *Alcyonium digitatum,* who outcompeted the early colonisers by over-growing. This seems to be the case at Hywind FOWF, which would indicate that the park is currently in the species-rich intermediate stage, moving towards a more *M. senile* dominating stage with less biodiversity. The taxonomical resolution in the data collected in 2018 limits the analysis of succession between the years. As in previous studies in the North Sea (De Mesel et al., 2015; Whomersley and Picken, 2003), a zonation was established just a few years after the installation of the structures. Echinoderms were present in high abundance and are considered an important grazer that affects the epifouling community (Witman, 1985) and could keep the epifouling colonisation growth suppressed.

*27. L250-251 I disagree, figure 8 shows one large increase and multiple small decreases, most with a lot of variation, in all but 1 case I see zero is included in the mean +- SD interval, indicating that the differences are likely not significant, although the authors did not test for this. I suggest to state that variation is large but no clear pattern was observed, or actually test for differences between years and present the results*

The section has been updated and statistical tests have been added.

L299-304 "Coverage of both hard and soft growth has significantly increased from 2018 to 2020 on both Suction Anchors and Mooring Lines, but not on the Turbine Substructures. The change in thickness is more variable compared to coverage, with a significant decrease of hard growth noted on both the Turbine Substructures and Suction Anchors, while an increase of the soft growth thickness was observed on the Turbine Substructures. Large standard deviations were observed for many of the measurements, due to the high variation between the structures. Further, the lack of lasers during the 2018 survey may have contributed to the variation of the measurements between the years."

*28. L273 please publish the data with the paper as online supplement or place it in a open access repository. Available upon request is not ideal for long term availability.*

A species list is now available as a supplementary table.

*29. L37 Whomersley and Picken is not a publication of OWF*

Reference removed

*30. L44 please define FOWF at first use*

Amended.

*31. L68 WROW should be WROV*

Amended.

*32. L72 what was the speed of the slower speed?*

0.2 knots, the text has been revised.

L85-86 reads "The three priority structures (HS01, HS02, and HS04) were investigated at a reduced speed of 0.2 knots (0.4 km/h), and at three sides (12 o'clock (north), 4 o'clock, and 8 o'clock) of the Turbine Substructures."

*33. L86 what is QC*

Quality control, the text has been revised.

*34. L87-89 what software did you use for the viewing and registration?*

The section has been revised and additional information has been added.

L101-106 "The analyses of the acquired video data were performed in two steps. The first step was analysed in real-time, from the live video feed from the WROV, and included documenting zonation, initial coverage estimates, and common species, which were registered into a field log template in Microsoft Word. During the second step, the video was played back using VLC Media Player and comprised quality control of the field logs as well as enumeration of individuals and assessment of percentage coverage of epifouling species. Lastly, the data was summarised into species lists, with separate lists for each structure and component."

*35. L224 Whomersley and Picken is not about OWF*

Amended.

266-267 "The high abundance of *M. senile* is consistent with findings from offshore structures in the North Sea (Whomersley and Picken, 2003; Kerckhof et al., 2012; De Mesel et al., 2015; Kerckhof et al., 2019)."

*36. L227 Wilhelmsson and Malm, 2008 is not in the North Sea.*

Reference removed.

*37. L290 Lengkeek is spelled wrong*

Amended.

**Reviewer 2**

1. *26-34 This an important issue and it would strengthen the point if more citations could be added here especially those from other than the industry*

The section has been revised and additional references have been added.

L29-38 now reads "The effects on local benthic habitats during installation works and operations of Offshore Wind Farms (OWF) are of complex nature and extend both below and above the surface of the sea. Previous studies have shown that OWFs can impact areas through the introduction and spread of alien species (De Mesel et al., 2015; Wilhelmsson and Malm, 2008), affect organic matter deposition (De Borger et al., 2021), and carbon assimilation (Mavraki et al., 2020), as well as alter community structures (Coates et al., 2014; Degraer et al., 2020; Hutchison et al., 2020; Wilhelmsson and Malm, 2008) through the loss of soft sediment habitats and the subsequent introduction of artificial hard bottom substrates. The newly created habitat is usually larger than the lost habitat (Wilson and Elliott, 2009). The recorded impacts also include recovery of the benthic biodiversity as a result of reduced trawling activities (Bergman et al., 2015; Coates et al., 2016) as well as an increase in nurseries for commercially important and/or protected species (Krone et al., 2017). The submerged structures (turbines and subcomponents on the seabed) introduce hard substrates into areas in which there were formerly lacking, thus facilitating colonisation."

*2. 74 – 75 please specify distance to the structure or give a least a range*

Distances have been added to the text.

L88-90 "A distance of approximately 0.5 m was maintained throughout the majority of the environmental survey and areas of interest were investigated at closer distances (<0.3 m). Occasionally, when sea state or obstructions occurred the distance to the structure was increased up to approximately 1 m."

*3. 68 misspelling of WROV*

Amended.

*4. 86 colonization refer to a process – shouldn´t it be more specific like "coverage" or "state of colonization?*

Updated to "initial coverage estimate".

*5. 93 please explain why nudibranchs and gastropods were excluded how did you overall selected your list and what was the rationale?*

The text has been revised. It was eggs of nudibranchs and gastropods that were excluded.

L113 "Eggs (from cephalopods, nudibranchs, and gastropods) identified during the survey were excluded from statistical analysis."

*6. 97-98 wasn´t the data secured such that the same person could compare and analyze the data later to insure more comparable data in 2018 and 2020?*

Unfortunately it was not feasible to have the 2018 data re-evaluated by the structural inspectors present during the 2020 survey.

*7. 104. I would like to see the species list! I failed to find it in results or reference to an appendix!*

A species list is now available as a supplementary table.

*8. 107. Same as above "different species of crustacean"!*

A species list is now available as a supplementary table.

*9. 111. I would think the term "percent coverage" is better*

Amended.

L157-158 "The coverage of epifouling taxa was found to be high (~80 % to 100 %), comprising predominantly species Metridium senile and Spirobranchus sp. across the majority of the turbine surfaces (Fig. 5)."

10. *112. Although -tidal depth ranges are well-defined terms I find it a little bit odd to use these depth categories in relation to floating structures where indeed the organisms do not experience the consequences of tides (e.g. the Kelp is not exposed to diurnal desiccation, changing light levels etc.) I Suggest to use depth ranges in meters instead*

The structures are anchored to the seabed using suction anchors and are attached to these by heavy chains. This results in an intertidal zone. However, intertidal zone is likely reduced as compared to a non-floating structure.

11. *130 Laminaria belong to the group of Phaeohyceae – should maybe be "Laminaria and other Phaeophyceae"*

Amended.

12. *144 "no significant differences were noted on the mooring lines…" This is a statement that should certainly be substantiated with additional information. What parameter did you look at (e.g. communities, coverage, biomass, individual species, biodiversity indices…)? What was your data format? test used … etc.*

"Significant" has been replaced by "clear".

13. *168-172. This part is not clear. The part about the coverage is clear but information on how you have distinguished between soft and hard fauna is missing (what species belongs to which groups) especially since the 2018 survey was performed by non-specialist. How was thickness determined from the underwater video film? Again bear in mind, that although this can easily be done by non-specialist, data is still subjective and the data quality could therefore be improved if the same person did both year (whether specialist or not). Legends on figure 7 an 8 missing information about the years and how changes is defied (2018 vs 2020?).*

This section has been revised accordingly. New figures have been introduced and now include a T-test, conducted on the coverage and thickness data. Details on specific taxonomic groups with regards to hard and soft growth, respectively, are now included.

L218-228: "Data from the 2018 inspection campaign, provided by REACH Subsea, was compared to the data acquired during the 2020 campaign (Table 2, Fig. 8). The coverage on the Turbine Substructures was not significantly different between the years, neither for the hard (P=0.82) nor for the soft growth (P=0.11). However, there was a significant decrease in the thickness of hard growth (P<0.001), whereas the soft growth increased in thickness (P=0.01). The coverage on the Suction Anchors increased in 2020 compared to 2018, both for the hard growth (P=0.002)

and soft growth (<0.001), whereas the thickness of the cover decreased, the change was significant for the hard growth (P<0.001), but not for the soft growth (P=0.10). For the Mooring Lines the coverage increased significantly both for the hard growth (P<0.001) and the soft growth (P<0.001). However, there were no significant changes in the thickness of the growth.

On the Turbine Substructures, the largest shift in composition was a loss of hydroids on 15 of 23 sub-components, and seven sub-components had a gain of macroalgae. On the Mooring Lines, there was a loss of hydroids on 61 of 125 sub-components and a loss of tubeworms on 49 sub-components, and a loss of barnacles on 45 sub-components."

14. *175. 175. Subheading should rather be "identification of species" as no non-indigenous species in fact was determined with certainty*

Amended.

15. *176. 176 – 178 should be under results!*

Amended.

16. *177. 193 - 197 This is a potential important finding, due to the threatened status of Desmophyllum, and I would recommend to contact external experts as you did with the Barnacles. A strong group I located just around the corner at Edinburg University.*

Advised experts agree that the colony is likely D. pertusum, but due to the small size and uncharacteristic appearance a positive identification would require close up imagery of the calyx using a stills camera. The text has been amended.

L257-260: "The shape of the colony tentatively identified as the deep-water coral D. pertusum is atypical for the species, however, similar dome-shaped colonies have been recorded on oil platforms in the North Sea (e.g., Gass and Roberts, 2006). Advised experts agree that the colony is likely D. pertusum, but due to the small size and uncharacteristic appearance a positive identification would require close up imagery of the calyx using a stills camera."

17. *178. 215-216. If diversity "is lacking" there is either 0 or 1 species present! Better to write whether there was fauna at all. Did you quantify species richness or diversity at all? In case this should be described under methods and results. Better to delete this sentence*

The sentence has been revised.

L272-273 "The painted Turbine Substructures harboured fewer taxa compared to the uncoated Mooring Lines."

18. *224 similar in what way?*

In regard to species composition and distribution. This text has been revised.

L282-283: "A depth zonation similar to, in regard to species composition and distribution, other offshore structures in the North Sea (Whomersley and Picken, 2003; Lengkeek and Bouma, 2009; De Mesel et al., 2015) was noted within the current survey area."

*19. 233. What do you mean by "not significant enough" if no statistical analysis was performed write instead "differences was not clear"*

Amended.

L292-9293: "The differences were not clear enough to indicate whether or not the currents or the distance to shore would affect the zonation and growth of epifaunal species."

*20. 235 Different I what way?*

The text has been revised.

L292-297: "The zonation noted along the Mooring Lines comprised a different species community than those identified at the Turbine Substructures. The Mooring Lines were generally dominated by M. senile and Balanoidea at the same water depths as where the Turbine Substructures were dominated by Spirobranchus sp. and M. senile. The top and upper-middle sections of the Mooring Lines were dominated by M. senile and Balanoidea. The middle chain comprised, overall, lower faunal colonisation."

*21. 239-241 How can you conclude the species succession follow the same trend as you have stated in the material and methods that only a few phyla was recorded by the non-specialists. Species succession implies that the relative contribution species to the community changes over time.*

The section has been revised, and together with the new section regarding changes in taxonomic composition between 2018 and 2020. We do believe that we can say that Hywind FOWF seems to follow the same pattern as other offshore structures in the North Sea. However, the limited taxonomical resolution should of course be considered.

L306-317: "The gain and loss of taxa observed indicates a shift in taxonomic composition between 2018 and 2020, with mainly a decrease in hydroids, tubeworms, and barnacles, this was corroborated in discussions with the survey team who performed the initial visual inspection in 2018, and they confirmed that faunal composition had changed between the two years, indicating a succession. The observed changes seem to follow the same trend regarding succession stages that has previously been observed on offshore installations in the North Sea (Rumes et al., 2013; Whomersley and Picken, 2003), tubeworms and hydroids have been reported as the first to colonise the structures, followed by M. senile and Alcyonium digitatum, who outcompeted the early colonisers by over-growing. This seems to be the case at Hywind FOWF, which would indicate that the park is currently in the species-rich intermediate stage, moving towards a more M. senile dominating stage with less biodiversity. The taxonomical resolution in the data collected in 2018 limits the analysis of succession between the years. As in previous studies in the North Sea (De Mesel et al., 2015; Whomersley and Picken, 2003), a zonation was established just a few years after the installation of the structures. Echinoderms

were present in high abundance and are considered an important grazer that affects the epifouling community (Witman, 1985) and could keep the epifouling colonisation growth suppressed."

22. *251 the information about the uncertain due to lack of consistent methodology should (as mentioned in the above several times) should be flagged up front in MS and it also concerns the above comments on succession.*

Information regarding inconsistencies between the years has been included in the Methodology section, as well as briefly in the Discussion section.